# Rhabdomyosarcoma of the Biliary Tract in Children: Analysis of Single Center Experience

**DOI:** 10.3390/cancers16173110

**Published:** 2024-09-09

**Authors:** Ewa Święszkowska, Dorota Broniszczak, Piotr Kaliciński, Marek Szymczak, Marek Stefanowicz, Wiesława Grajkowska, Bożenna Dembowska-Bagińska

**Affiliations:** 1Department of Oncology, The Children’s Memorial Health Institute, Al. Dzieci Polskich 20, 04-730 Warsaw, Poland; e.swieszkowska@ipczd.pl (E.Ś.); b.dembowska@ipczd.pl (B.D.-B.); 2Department of Pediatric Surgery and Organ Transplantation, The Children’s Memorial Health Institute, Al. Dzieci Polskich 20, 04-730 Warsaw, Poland; p.kalicinski@ipczd.pl (P.K.); m.szymczak@ipczd.pl (M.S.); m.stefanowicz@ipczd.pl (M.S.); 3Department of Pathology, The Children’s Memorial Health Institute, Al. Dzieci Polskich 20, 04-730 Warsaw, Poland; w.grajkowska@ipczd.pl

**Keywords:** rhabdomyosarcoma, biliary tract, liver, chemotherapy, radiotherapy, surgery, transplantation

## Abstract

**Simple Summary:**

Rhabdomyosarcoma of the extrahepatic and intrahepatic bile ducts is a rare location in children, and it poses a significant challenge for both oncologists and surgeons. Key elements of treatment where there are no clear guidelines include surgical treatment of RMS of the bile ducts, chemotherapy and radiotherapy. The authors wish to share the experience of a highly specialized center that has access to a wide range of therapeutic and treatment options, including the management approach and the outcomes achieved. We review cases treated at our center, discuss treatment approaches and outcomes and compare results with the existing literature and guidelines.

**Abstract:**

Rhabdomyosarcoma (RMS) of the biliary tract is a rare tumor in children, constituting 0.5–0.8% of all pediatric RMS. Still, it is the most common malignancy in this location in children. Due to its rarity and location, it may cause diagnostic and treatment difficulties. Above all, there are no therapeutic guidelines specific for this tumor location. The aim of the study was to present an analysis of our experience with the treatment of children with biliary tract rhabdomyosarcoma (RMS) and discuss clinical recommendations for this specific location published in the literature. A retrospective analysis of medical records of eight children with biliary tree RMS treated in one center between 1996–2022 was performed. Records of eight children, five boys and three girls aged 2 yrs 6 mo to 16 yrs 9 mo (median—6 yrs) were analyzed. All patients presented with jaundice as the first symptom. In two patients, initial diagnosis of a tumor was established. For the remaining six, the primary diagnoses were as follows: choledochal cyst—one, malformation of the biliary ducts—one, choledocholithiasis—one, cholangitis—three. In four patients, the extrahepatic bile ducts were involved; in four patients, both the intrahepatic and extrahepatic bile ducts were involved. Embryonal RMS was diagnosed in seven patients (three botryoides type). Alveolar RMS was found in one patient. Biopsy (three surgical, four during endoscopic retrograde cholangiopancreatography (ERCP)) was performed in seven patients. One child underwent primary partial tumor resection (R2). Seven patients received neoadjuvant chemotherapy, followed by delayed resection in five, including liver transplantation in one (five were R0). Two patients did not undergo surgery. Radiotherapy was administered in four patients (two in first-line treatment, two at relapse/progression). Six patients (75%) are alive with no evidence of disease, with follow-up ranging from 1.2 yrs to 27 yrs (median 11 yrs. and 4 mo.). Two patients died from disease, 2 y 9 mo and 3 y 7 mo from diagnosis. Children presenting with obstructive jaundice should be evaluated for biliary tract RMS. The treatment strategy should include biopsy and preoperative chemotherapy, followed by tumor resection and radiotherapy for residual disease and in case of relapse.

## 1. Introduction

The most common sites of rhabdomyosarcoma (RMS) in children are the head, genitourinary tract and extremities. The biliary tree is considered a very rare location of this tumor type, accounting for less than 1.5% of all RMS [1,2,3]. Biliary tract RMS is typically of embryonal histology, though alveolar type has been also described [1,2,3]. Due to its rarity, the very common symptom obstructive jaundice may be taken for another entity, such as choledochal cyst, choledocholithiasis, malformation of the bile ducts or cholangitis, delaying proper diagnosis [4,5,6,7]. Treatment of biliary tract RMS may be challenging in terms of carrying out radical surgery and radiotherapy as recommended for other RMS locations and risk groups [8,9,10]. The approach to the surgical treatment of RMS has changed over 25 years. In the past, these patients were primarily operated on, often undergoing extrahepatic biliary duct resection. Primary radical resection was frequently impossible, leaving gross residual tumor. However, the incorporation of chemotherapy showed, during subsequent exploratory operations, the absence of residual disease, including local infiltrations beyond the biliary tract. The high chemosensitivity of RMS results in tumor shrinkage, providing conditions for safe resections and cure even in locally advanced disease. The treatment of children with rhabdomyosarcoma includes a combination of chemotherapy, surgery and radiotherapy. Chemotherapy in the neoadjuvant and adjuvant setting has an established role in the treatment of RMS. There are no specific protocols for biliary tree RMS, but treatment strategy is as in other locations. There are data supporting the benefit of aggressive surgery, but it is not so clear for radiotherapy, given the young age of patients with biliary tree RMS and the long-term sequelae of irradiation [11,12,13,14,15,16]. The role of liver transplantation for RMS in patients with intrahepatic infiltration is not clearly defined. Our study was done with the aim to analyze our experience and results of the treatment of children with biliary tract rhabdomyosarcoma, a very uncommon tumor in children, and discuss specific recommendations based on the latest literature.

## 2. Materials and Methods

Among 227 with RMS treated between 1996–2022 in our center, 8 patients (3.5%) were diagnosed with biliary tract RMS. Their medical records were reviewed and analyzed for sex, age at diagnosis, symptoms, location (extra, intrahepatic), tumor size, histological subtype, disease stage, response to chemotherapy, type of surgery (primary/delayed, type of resective surgery and its radicality), other treatment options and outcome. Disease stage was assessed by diagnostic imaging (US and CT or MRI) and response to chemotherapy was assessed with the same diagnostic tools. To establish a histological diagnosis, the first step was an attempt at endoscopic biopsy sampling. In the event of unsuccessful sampling or obtaining non-diagnostic material, patients were considered for material collection through laparoscopy/laparotomy access. All patients were treated according to the protocol involving a combination of chemotherapy, surgery and, in selected cases, radiation. Since there are no specific chemotherapy protocols for rhabdomyosarcoma of the biliary tract in children, chemotherapy consisting of anti-cancer drugs as for other RMS locations was implemented. Treatment decisions were made by a multidisciplinary team of pediatric oncologists and pediatric surgeons. Response to chemotherapy was assessed by imaging studies. Complete response (CR) was defined as no residual tumor on radiological assessment after neoadjuvant chemotherapy or no viable cells present in tumor specimen after delayed resection; good response (GR) was defined as two thirds regression of tumor size, poor response (PR) as a regression between one third and two thirds, objective response (OR) as a regression of less than one third, and progressive disease (PD) was defined as increase in tumor size or new metastatic lesions. Surgical procedures were defined as primary resections if performed before the administration of chemotherapy or delayed if conducted after chemotherapy administration. Types of surgery performed were defined as limited (A) when local tumor excision or resection of extrahepatic bile ducts without resection of hepatic ducts or surrounding organs were performed and extended surgery (B) when resection of extra and partly intrahepatic bile ducts, resection of parts of the hepatic ducts and/or liver resection or pancreaticoduodenectomy with resection of extrahepatic bile ducts or liver transplantation were performed. Resection status was described as follows: complete surgical resection—no viable tumor cells were found on pathological examination (R0), microscopically incomplete surgical resection (R1) or gross tumor residual (R2). Due to the small number of patients, no statistical methods, including regression analysis, were applied. The publication is a retrospective case series describing a group of pediatric patients with biliary tract RMS.

## 3. Results

### 3.1. Patients’ Demographics

There were five boys and three girls, aged from 2.6 to 16 years 9 months (median 6 years) at diagnosis. Two patients were over 10 years of age.

### 3.2. Clinical and Laboratory Data

All patients presented with jaundice and abdominal pain as first symptoms. Conjugated bilirubin concentration ranged from 1.65 to 11.3 mg/dL (median 5.86 mg/dL). All patients had increased activity of transaminases, from 2 to 4.5 times that of normal values (AST ranged from 1.3 to 4.5 times, ALT 1.1–4.2 times). GGTP ranged from 1.6 to 54 times that of normal value). Two patients were suspected to have a tumor before admission to our hospital. For the others, the diagnoses were as follows: choledocholithiasis in one, malformation of the bile ducts in one, choledochal cyst in one, and cholangitis in the remaining three patients.

### 3.3. Tumor Extension

In four patients, tumor was located in the extrahepatic bile ducts. Four children presented both extra and intrahepatic involvement. In six patients, tumors were smaller than 5 cm; in two, the diameter of the tumor was 9 and 12 cm. One patient had metastatic disease at diagnosis—neoplastic process involved the peritoneum, ileum and liver surface.

### 3.4. Histopathological Diagnosis

In seven children, embryonal RMS (ERMS) was diagnosed, including three with botryoides subtype and alveolar RMS (ARMS) in one.

### 3.5. Chemotherapy

Seven patients received neoadjuvant chemotherapy. Preoperative chemotherapy consisting of CAV (cyclophosphamide, doxorubicin, vincristine) alternating with IE (ifosfamide, etoposide) and IF/ADM (ifosfamide, doxorubicin) was administered to four patients. The remaining three children received IVA (ifosfamide, vincristine, dactinomycin) chemotherapy. Seven patients received four to nine courses of preoperative chemotherapy (median—six).

### 3.6. Response to Neoadjuvant Chemotherapy

One patient had complete response (CR) on imaging studies after preoperative chemotherapy, three patients had good response (GR), one patient had poor response (PR), one patient had objective response (OR) and one patient had progressive disease (PD). Five patients received adjuvant chemotherapy. The one patient who had primary surgical resection received adjuvant chemotherapy after radiotherapy, consisting of VAC, IVA, VCR/ADM and ETIF.

### 3.7. Surgical Procedures and Tumor Resection Status

Primary resection of the tumor was performed in one patient. This patient had an initial diagnosis of choledochal cyst. During surgery, it was found to be a bile duct tumor. The resection was performed without healthy tissue margins (R2). Seven remaining patients underwent biopsy: in two patients, biopsy was taken through an open procedure; in one case, the procedure was performed laparoscopically; in four, biopsy was taken during endoscopic retrograde cholangiopancreatography (ERCP).

Delayed resection was performed in five patients. Three patients had their extrahepatic bile ducts removed and reconstructed using a Roux-en-Y loop. All were R0. In one patient, during surgery, a tumor of 4–5 cm surrounding the cystic duct was identified, and only the gallbladder and cystic duct were removed. It was also assessed as R0. In one patient, a tumor involving both extra- and intrahepatic bile ducts was present on imaging studies despite chemotherapy. This patient was qualified for simultaneous pancreaticoduodenectomy (Whipple’s procedure), total hepatectomy and liver transplantation from a living donor. Despite extensive surgeries, no surgical complications were identified in our patients.

### 3.8. Radiation Therapy

External beam radiation was implemented in the first line treatment in two patients: the one with primary R2 resection and the other who was never operated on. The total radiation dose was 45 Gy to the tumor/tumor bed with margins. Two other patients received radiotherapy for disease relapse.

### 3.9. Relapse/Progression

One patient (Pt. No. 3) relapsed 24 months from diagnosis. He received second-line chemotherapy and radiotherapy. He is alive and disease-free 23 years from diagnosis. Two children (Pts. No. 4 and 5) had progressive disease 28 and 24 months from diagnosis. One of them (Pt. No. 4) received chemotherapy, then underwent surgery and radiotherapy. The other one (Pt. No. 5), who was already irradiated, received chemotherapy. Both patients died (3 yrs 7 mos and 2 yrs 9 mos from diagnosis).

### 3.10. Outcome

Six out of eight patients (75%) are alive, with follow-up ranging from 1.2 years to 27 years (median 11 yrs and 4 mos). The patients’ characteristics are presented in Table 1 and Table 2.

## 4. Discussion

The main limitation of research on RMS of the biliary tract in children is the small number of patients, even when considering multicenter analyses. Therefore, any single center experience may add some information contributing to the discussions about the best treatment for children with this tumor [1,2,3,11,17].

### 4.1. Epidemiology

RMS of the biliary tree is a rare entity. Available published data indicate that it accounts for 0.5 to 1.5% of all RMS in children [1,2,3]. Our series showing 3.5% of such patients among all RMS children is overestimated, coming from a single center where many patients with liver malignancies are referred. Biliary tree RMS, according to the literature, is a disease of young children, around 3 years of age [3]. Our results are not in line with these data, since the median age of our patients was 6 years. We report a higher percentage of boys, as in most case series. RMS in the adult population is exceedingly rare. To our knowledge, there is only a single published case report describing patients with biliary tree RMS [18].

### 4.2. Diagnosis of Biliary Tract RMS

Due to the rarity of biliary RMS, its most common symptom—obstructive jaundice—may be taken for other entities such as choledochal malformation, choledocholithiasis, cholangitis or pancreatic tumor. In our series, only two patients had an upfront diagnosis of a tumor; for the remaining patients, other conditions were misidentified initially. The suspicion of a bile duct tumor was eventually raised in all these children after admission to our center. This issue is also raised by other authors [4,5]. One of our patients diagnosed with gallstone disease underwent laparoscopic cholecystectomy. She was treated for persistent obstructive jaundice for 2 months before the correct diagnosis was made. Another patient was diagnosed with a bile duct cyst, and thus, she was qualified for choledochal cyst resection surgery. When retrospectively analyzing imaging studies of this girl, we found that despite both ultrasound and computer tomography being strongly suggestive of a bile duct cyst, scintigraphy did not provide such confirmation, failing to show a clear image of the cyst (Figure 1 and Figure 2). Such clinical situations may lead to the performance of primary resection, which is rarely a complete one (R0) [1,7,11]. Thus, in the differential diagnosis of bile duct cysts and obstructive jaundice in children, biliary RMS must always be considered. All our patients manifested obstructive jaundice as the first symptom of disease, which was reported in 60–80% of patients in other studies [3,4,19].

Biliary tract RMS can originate from anywhere along the biliary tree: the intra- or extrahepatic bile ducts, the gallbladder, or the hepatopancreatic junction (ampulla of Vater) [3,7,20,21,22,23]. RMS very rarely presents as a liver tumor [1,11]. In our group, in one patient, RMS was located in the cystic duct. In another, it extended towards the duodenum, while in the third patient, on diagnostic imaging, tumor was present within the liver, extrahepatic biliary tree and pancreas.

Due to the rarity of biliary tract RMS and its presentation with non-specific symptoms like jaundice and abdominal pain, early diagnosis is often challenging. The intricate anatomy of the biliary system complicates imaging interpretation and biopsy procedures, leading to delays in accurate diagnosis. Although the authors do not have experience in using positron emission tomography (PET) in the diagnosis of biliary tract RMS (PET was not available at our center when the described patients were being treated), it appears, based on available case reports where this diagnostic technique was used, that it could be a valuable tool for distinguishing biliary tract RMS from bile duct cysts or other entities in this site [24,25]. The use of PET has been already applied in the diagnosis of biliary tract cancers and their metastases in adults. PET scans utilize a radioactive tracer, typically fluorodeoxyglucose (FDG), which accumulates in tissues with high metabolic activity, like malignant tumors. RMS, being a highly aggressive and metabolically active tumor, usually shows increased FDG uptake on PET scans, leading to a “hot spot” appearance. The possibility of performing PET was introduced at our center in 2022, and the last patient presented in our study had a suspected finding in MRI after surgery monitored using this technique.

Regional and distant metastases have been reported in 30–40% patients at diagnosis [1,21]. None of our patients had distant metastases at diagnosis, and local spread was observed in one.

Histological diagnosis is established based on tissue samples obtained through primary tumor resection, open/laparoscopic biopsy or biopsy performed during ERCP. Half of our patients had histological diagnosis with ERCP without any complications. Laparoscopy was performed in two cases; however, in one instance, the procedure had to be converted to a laparotomy. Laparoscopically, the abdominal cavity can be assessed for metastases and tumor infiltrations, which may not always be visible on imaging studies. The use of ERCP for tumor tissue collection and laparoscopy allows avoiding invasive procedures. In our series, ERCP proved to be a safe method for both diagnosis and decompression of the biliary tract (Figure 3). From the published series, about 12% of patients had diagnostic ERCP [1]. In our opinion, it should be more widely used in establishing a diagnosis of biliary RMS.

Two main histological subtypes of RMS, embryonal and alveolar, are distinguished, embryonal being the most common. The alveolar subtype constitutes about 2% of RMS in this location [1]. In our series, all but one patient had ERMS, and three were botryoid type, a subset of ERMS.

### 4.3. Treatment

The standard of care for rhabdomyosarcoma of the biliary tract requires multimodality treatment, including chemotherapy, surgical resection, and/or radiation therapy. The standard chemotherapy protocols combine vincristine, actinomycin, and cyclophosphamide/ifosfamide. Surgery aims for complete tumor resection, which can be challenging due to the tumor’s location in the biliary tract. Radiation therapy may be employed postoperatively to target residual disease, especially if complete surgical resection is not achievable. The protocol is often adapted based on the tumor’s stage, histological subtype, and patient factors, with ongoing clinical trials helping refine these strategies. Multidisciplinary teams are crucial in tailoring treatment to optimize outcomes while minimizing complications.

#### 4.3.1. Surgery and Neoadjuvant Chemotherapy

The role of biliary RMS resection and its extent has been discussed by some authors, but this discussion seems to apply to patients who did not receive neoadjuvant chemotherapy [2,17]. In most reports, surgery was usually performed after preoperative chemotherapy, and often no disease on pathology was found. In most patients, thickening of the bile duct walls is still present on imaging after chemotherapy. Even if it is not, surgical excision should be performed whenever possible to achieve the highest degree of radicality. The risk of mortality associated with recurrence of RMS is higher than that of the surgery itself, even in the absence of tumor cells in the resected specimen [1].

As for primary surgery, it should only be performed to establish diagnosis and possibly determine the extent of the disease if it cannot be done by other methods (ERCP, MRCP, etc.).

Some reports demonstrate that chemotherapy alone leads to complete remission of RMS [1]. Consequently, the necessity of surgical treatment was questioned. From the available literature, there are reports which demonstrate that there are patients with biliary RMS cured with chemotherapy alone. Among seventeen patients described by Urla et al. there were five patients who were not operated but received chemotherapy (according to CWS 96 protocol) alone. Among these patients, three were alive without evidence of disease at the time of the publication [2]. In Perrucio’s et al. study, there is also one patient treated with VAIA chemotherapy alone who is alive and disease-free 140 months from diagnosis [19]. Spunt and colleagues reported on eight patients treated with chemotherapy alone who are long-term survivors [17]. In our series, two patients who were treated with chemotherapy alone died; one of them had ARMS. These observations are inconclusive but suggest that this method alone can be efficient for cure in some patients. 

In systematic review and meta-analysis of biliary rhabdomyosarcoma in children published in 2021, surgery after initial chemotherapy was recommended whenever possible [1]. It was shown that the lack of tumor resection is an independent risk factor for death and is significantly associated with relapse. Series published by Guerin and Urla confirm it [2,11]. It was also observed in our series.

The significant diversity of locations and the extent of the tumor in the biliary ducts suggest performing various surgical modifications, ranging from cholecystectomy through the excision of extrahepatic bile ducts with reconstruction of the bile ducts with a Roux-en-Y loop, to pancreatoduodenectomy and complete hepatectomy with liver transplantation. The collected and published data demonstrate that one standard of treatment is difficult to establish in children with biliary RMS; rather, a therapeutic plan should be prepared individually for each patient [2,3,11,17,19] (Table 3).

#### 4.3.2. Liver Transplantation

In 2016, we performed a total liver resection in intrahepatic, extrahepatic and intrapancreatic rhabdomyosarcoma (RMS) and simultaneous liver transplantation with pancreaticoduodenectomy in a patient with such disease extent. No RMS cells were detected in the explanted organs of this patient. The patient is alive 7 years from diagnosis. The decision for liver transplantation was based on persistent tumor presence in all involved locations on MRI studies after multiple courses of chemotherapy and supported by a single report in the literature (Figure 4) [9]. The authors did not find viable tumor cells on pathology in the right lobe, but they were present in the left one. They speculate whether this information might have led to a change in their decision to limit the resection to the left lobe. Based on computer tomography or magnetic resonance imaging, it is not possible to differentiate whether the visible changes are only fibrosis or residual disease. Perhaps positron emission tomography could facilitate the differentiation of such changes [16,24]. Shortly after our case, another case report emerged of a patient who underwent pancreaticoduodenectomy for biliary tract RMS and about 3 years later underwent liver transplantation due to metastases in the liver [10]. The follow-up at the time of publication was 6 months without major complications or tumor recurrence [10].

The cited cases confirm the necessity of both appropriate chemotherapy and aggressive surgery in selected patients [8,9,24]. In the publication by Guerin et al. [11], there was a patient who underwent liver rescue transplantation after a relapse, with poor results. In our experience, performing liver transplantation due to a relapse of any liver tumor has almost always resulted in recurrence. The performance of the Whipple procedure along with liver transplantation in different orders (before, simultaneously and after liver transplantation) plays a role in the treatment of cholangiocarcinoma, pancreatic neuroendocrine tumors and adenocarcinoma with metastases to the liver in adults [8,10].

#### 4.3.3. Adjuvant Chemotherapy

Adjuvant chemotherapy after delayed surgery is recommended according to risk-adapted protocols, as in other RMS sites, and is not specific to the biliary tract location. The available literature does not depict nor suggest the number of chemotherapy courses given after delayed surgery [2,3,11,17,19]. Usually, it follows protocols using a risk classification scheme combining data about the clinical group and disease stage. In our series, we applied a median of four postoperative courses.

#### 4.3.4. Role of Radiotherapy

RMS are radiation-sensitive tumors. It is recommended in cases of incomplete tumor resection, alveolar histology and at relapse [1,2]. Aye et al. report that 88% of patients who underwent radiotherapy, including those with micro and gross residual disease, did not experience recurrence, while patients who did not receive radiotherapy had local relapses [3]. In Guerin’s study, four patients did not undergo surgery but instead received radiotherapy only as local treatment [11]. Among these patients, one experienced a relapse. They also observed that patients who underwent surgery and radiation had an 11% relapse rate, compared to 27% in those treated surgically (no statistical significance). Radiation therapy, according to the publications, contributes to reduced mortality in patients with biliary RMS [1].

However, radiotherapy carries a risk of early and late complications [1,2,8]. Complications related to radiotherapy can be significant, especially in the younger patient population. These complications may include veno-occlusive disease, nodular regenerative hyperplasia, hepatic dysfunction and secondary tumors.

In our series, radiotherapy was implemented in two patients as first-line therapy and in two at relapse, resulting in the cure of two patients, one from each group. The two irradiated patients who are long-term survivors, as for now, do not present late effects of radiotherapy.

#### 4.3.5. Novel Treatments

New treatment strategies, such as introduction of molecular targeted drugs and immunotherapies, have shown superior efficacy and beneficial clinical outcomes as compared to standard treatments in selected childhood malignancies such as neuroblastoma, non-Hodgkin lymphoma, low-grade gliomas and melanoma. However, benefits of new approaches in pediatric RMS are not yet available. Genomic profiling of childhood soft tissue sarcoma, which will eventually help to elucidate and discover clinically meaningful biomarkers, as of today is not standardly performed. Studies on the role of tumor mutational burden and its role as a biomarker in predicting response of soft tissue sarcomas to immune checkpoint inhibitors are inconclusive, since they have a low mutational tumor burden, and the results are based on small sample populations. Further studies of biomarkers predicting the response to treatment and identification of druggable molecular targets are needed [26]. In our material, molecular studies were not carried out. 

### 4.4. Patients with Metastases at Diagnosis and Relapses

Regional and distant metastases have been reported in 30–40% of patients at diagnosis [1,21]. Patients with localized disease have better survival than those with metastases, although the mortality is not uniform in cases of embryonal metastatic RMS.

Urla et al. described five patients with distant metastases. In two of them, metastases disappeared after chemotherapy, while one had lung metastases surgically removed. Two patients died due to disease progression. In their material, survival was 40% in the metastatic group, compared to 63% in patients with localized disease; it was not statistically significant [2]. It was reported that patients with IRS Group IV (metastatic disease) ERMS have 5-year failure-free survival (FFS) of about 30% [1].

Recurrence is observed in about 30% of patients, with a mortality rate of approximately 80% linked to the relapse, as indicated by Fuchs [1], showing that recurrence of RMS has a poor prognosis. Therefore, the necessity of surgery in cases of complete tumor remission after chemotherapy seems to be justified considering the high risk of relapses, even when there is no viable tumor in the pathological examination. Such observations were cited by Guerin et al. [11]: among their patients, one of the six who had no viable tumor cells in the pathological specimen experienced a relapse. In our data, one of the two relapses also occurred in a patient who had no viable tumor in the histopathological specimen. The recurrence in this patient was treated with chemotherapy and radiotherapy without surgery, and the patient is a long-term survivor, free of disease. On the other hand, the analysis of risk factors associated with recurrence conducted by Guerin and colleagues, who assessed factors such as nodal status, type of resection (R0 vs. R1 and R2), tumor size, use of radiotherapy and timing of surgery (primary vs. delayed), showed that the only statistically significant factor for recurrence was the tumor size at the time of diagnosis. This observation is not consistent with our experiences, where two patients with tumors measuring 9 and 12 cm are alive with a long observation period.

### 4.5. Outcome

The survival rates for patients with RMS range from 85–89% for patients with low disease stage and favorable histology to 0–30% for patients with alveolar histology and distant metastases, with an average between 55 and 77% for all RMS patients [2,3,7,11,12,13,14,15,16,17,19,20,21,22,27] (Table 3). Analysis of available publications indicates that the overall survival rate of patients with biliary RMS is 65% [1]. It is noteworthy that even in the group of patients with Stage IV IRS Group IV (metastatic disease), 5-year failure-free survival (FFS) was 32% for ERMS. The alveolar subtype is always associated with poor prognosis. This is consistent with our observations, though, based on one patient with ARMS only. The girl was 16 years old at the time of diagnosis, which is more common for this RMS subtype [1,2]. At the time of diagnosis, this patient had locally disseminated disease, and despite an initial good response to chemotherapy, the treatment ended with failure.

Fuchs et al. in 2021 analyzed available publications (65 studies) on biliary tract rhabdomyosarcoma (RMS), with the majority being case reports, 12 case series, and five study groups (a total of 176 patients). Data from these 176 biliary tree RMS patients, as well as from a small series of patients including ours, show that long-term overall survival can be achieved in over 65% of patients. The worst prognosis is among patients with alveolar histology [1,2,19]. None of the patients with alveolar histology from CWS, Italian and our study survived [1,2,19].

An analysis comparing patients who underwent primary surgery showed that resection within healthy tissues is rarely possible (about 25% of cases) [1]. Owing to chemotherapy and radiotherapy, the survival of these patients is quite good. Resections performed after neoadjuvant chemotherapy provide a 60% rate of patients with tumor-free resection margins. In Fuchs’ analysis, a definitive answer was also found regarding treatment outcomes in cases of no surgical resection. The mortality in these cases was 63%, and the absence of surgery was a statistically significant factor for relapse and death due to RMS. Even though our study has limitations, due to the small number of patients coming from one institution treated during a long span of time, our observations add more data to this specific group of patients and may further support particular treatment recommendations [1,2,19].

## 5. Summary

Neoadjuvant chemotherapy resulted in CR and GR allowing for safe radical tumor resection, leaving radiotherapy for incompletely resected tumors and relapses. The rare occurrence of biliary tree RMS should oblige that treatment be carried out in centers experienced in hepatobiliary pediatric surgery and pediatric oncology. The individual patients documented in the literature who achieved cure solely with chemotherapy, without any local treatment, should be regarded as a significant contribution to the ongoing discussion regarding which children, if any, could be considered candidates for such conservative therapeutic approaches. However, recent analyses strongly contradict such a course of action and underscore the pivotal role of surgical resection in treating RMS. The removal of the entire liver followed by transplantation, along with the pancreaticoduodenectomy procedure, has a role in the treatment of rhabdomyosarcoma (RMS) of the biliary tract. With advancements in surgical techniques and chemotherapy, patients with such an advanced stage of the disease may achieve favorable long-term outcomes.

## 6. Conclusions

Diagnostic challenges associated with RMS of the biliary tract result from the diversity of the disease’s forms and the ambiguous clinical presentation. Therapeutic difficulties often arise due to the unavailability of radical surgical treatment before implementing combined approaches. The standard of care and recommended treatment strategy for rhabdomyosarcoma of the biliary tract requires multimodality treatment, including chemotherapy, surgical resection, and/or radiation therapy. The standard chemotherapy protocols combine vincristine, actinomycin and cyclophosphamide/ifosfamide. In exceptional cases, such as RMS involving intrahepatic and intrapancreatic bile ducts, consideration may be given to liver transplantation and pancreaticoduodenectomy (Whipple procedure) as a treatment option.

## Figures and Tables

**Figure 1 cancers-16-03110-f001:**
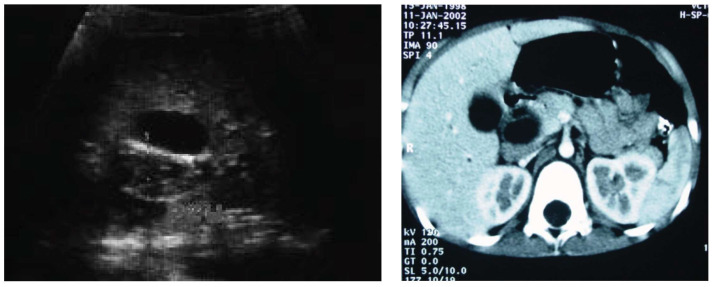
Ultrasound and computed tomography in a patient with biliary tract RMS who was incorrectly diagnosed with a bile duct cyst—cystic dilation of the common bile duct up to 2 cm, filled with thick bile, and dilation of intrahepatic bile ducts.

**Figure 2 cancers-16-03110-f002:**
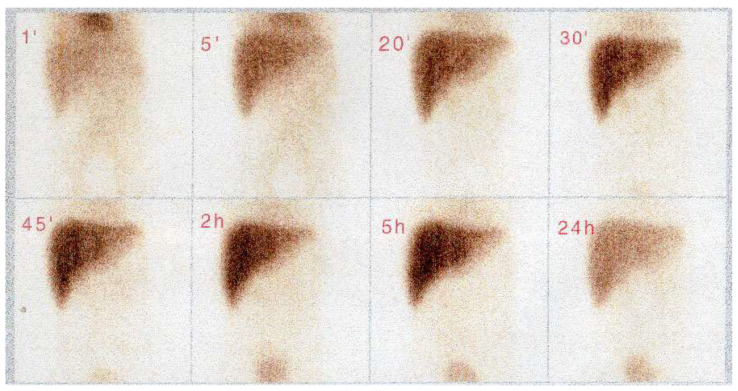
Scintigraphy in a patient with biliary tract RMS who was incorrectly diagnosed with a bile duct cyst—no bile passage to the intestines, impaired liver cell function and no cystic dilation seen in the liver hilum with tracer accumulation.

**Figure 3 cancers-16-03110-f003:**
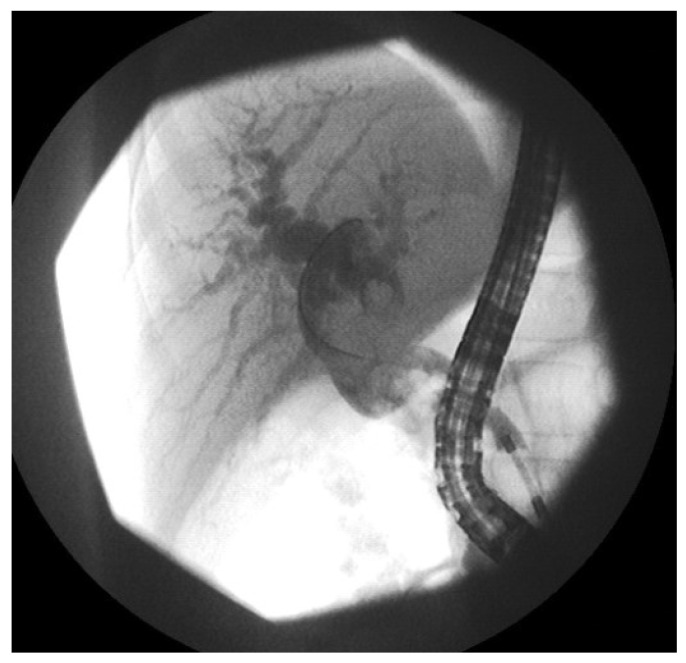
Endoscopic Retrograde Cholangiopancreatography in a patient with suspected biliary tract RMS. During this procedure, tissue samples were collected for histopathological examination.

**Figure 4 cancers-16-03110-f004:**
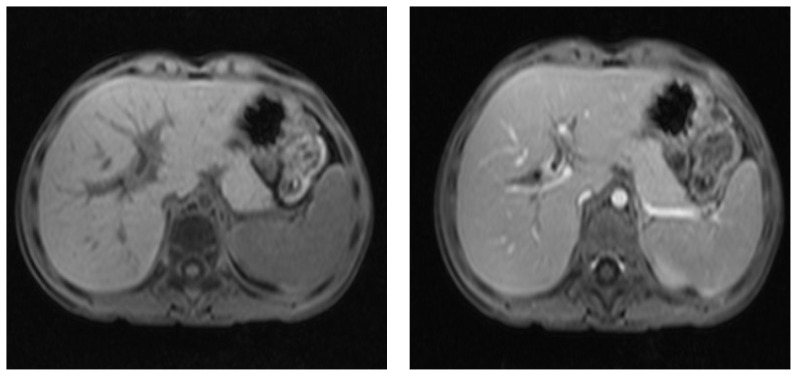
Magnetic resonance imaging of a patient with intrahepatic and extrahepatic RMS showing persistent intrahepatic, extrahepatic, and pancreatic head infiltrations despite chemotherapy. The patient underwent a pancreatoduodenectomy and complete hepatectomy with liver transplantation.

**Table 1 cancers-16-03110-t001:** Patients’ characteristics.

Pt No. Year of Treatment	Sex	Age (Years)	Tumor Diameter (cm)	Location	Upfront Surgery	Pathology	Neoadjuvant Chemotherapy	Response to Chemotherapy on Imaging Studies	Delayed Surgery	Adjuvant Chemotherapy	Radiotherapy	Outcome	Follow-Up
1. 1996 D.T.	M	9.2	9	Extrahepatic biliary tract	Open biopsy	ERMS	CAV/IE	GR	A-1, R0	CAV/IE	No	Alive in first CR	26 years from diagnosis
2. 2002 H.A.	F	4.2	12	Extrahepatic biliary tract	Partial resection, A2, R2	ERMS (botryoides)	No	Not applicable	No	VAC, IVA, VCR/ADM, IE	Yes 4500 cG/tu (First line)	Alive in first CR	21.7 years from diagnosis
3. 2004 L.Ł.	M	5.6	3	Extrahepatic biliary tract	ERCP biopsy	ERMS (botryoides)	CAV/IE	PR	A-2, R0	No	Yes 5040 cG/tu (Second line, at relapse)	Relapse local 24 months from diagnosis. Treatment (radiotherapy, chemotherapy)	Alive in second CR (22.9 years from diagnosis)
4. 2011 L.M.	F	16.9	3.7	Extrahepatic biliary tract	Laparoscopicbiopsy	ARMS, metastatic disease	IVADo, VAC	GR	No	No	Yes 5040 cGg/tu (At relapse)	Progression of disease. 28 months from diagnosis Treatment (radiotherapy, chemotherapy)	DOD (3 y 7 mo from diagnosis)
5. 2013 L.K.	M	2.6	3 lesions 1.8 1.7 1.6	Intrahepatic and Extrahepatic biliary tract	ERCP biopsy	ERMS (botryoides)	CAV/IE, VAC	PD	No	Yes	Yes 4500 cG/tu	Progression of disease 24 months from diagnosis Treatment (chemotherapy and chemotherapy)	DOD (2 y 9 mofrom diagnosis)
6. 2015 P.M.	F	12.9	3.7	Intrahepatic and Extrahepatic biliary tract	ERCP biopsy	ERMS	CAV/IE	OR	B-3,B-4, R0	CYVADIC	No	Alive in first CR,	7.1 years from diagnosis
7. 2020 R.P.	M	6.8	3.2	Intrahepatic and Extrahepatic biliary tract	ERCP biopsy	ERMS	CWS (IVA)	CR	A-2, R0	Vinorelbine/ CTX	No	Alive in first CR,	3.3 years from diagnosis
8. 2022 S.I.	M	2.6	4.7	Intrahepatic and Extrahepatic biliary tract	Open biopsy	ERMS	CWS (IVA)	GR	A-2, R0	Vinorelbine/ CTX	No	Alive in first CR	1.2 years from diagnosis

CAV—Cyclophosphamide, doxorubicin, vincristine; IE—etoposide, ifosfamide; IF/ADM—ifosfamide, doxorubicin, IVADo—ifosfamide, vincristine, dactinomycin, doxorubicin; VAC—vincristine, dactinomycin, cyclophosphamide; IVA—ifosfamide, vincristine, dactinomycin; Vinorelbine/CTX—vinorelbine, cycklophopshamide; CR—complete response; GR—good response; PR—poor response; OR—objective response; PD—progressive disease; DOD—died of disease; Complete surgical resection—no viable tumor cells were found on pathological examination (R0), microscopically incomplete surgical resection (R1), gross tumor residual (R2); ERCP—endoscopic retrograde cholangiopacreatography; ERMS—embryonal rhabdomyosarcoma; ARMS—alveolar rhabdomyosarcoma.

**Table 2 cancers-16-03110-t002:** Summary of patients’ characteristics.

	Number of Patients	%
**Gender**		
male	5	62.5%
female	3	37.5%
**Age at diagnosis**		
<10 years old	6	75%
>10 years old	2	25%
**Symptoms**		
jaundice	8	100%
abdominal pain	8	100%
**Suspected diagnosis at presentation**		
tumor	2	25%
choledochal cyst	1	12.5%
malformation	1	12.5%
choledocholithiasis	1	12.5%
cholangitis	3	37.5%
**Location**		
Extrahepatic	4	50%
Both extra and intrahepatic	4	50%
**Histology**		
embrional RMS	7	87.5%
-botrioid embrional RMS	3	
alveolar RMS	1	12.5%
**Tumor size**		
<5 cm	6	75%
>5 cm	2	25%
**Metastases**		
no	7	87.5%
yes	1	12.5%
**Biopsy at diagnosis**		
yes	7	87.5%
-ERCP	4	
-Laparoscopy/laparotomy	3	
no	1	12.5%
**Neoadiuvant chemotherapy**		
yes	7	87.5%
no	1	12.5%
**Primary surgery**		
-biopsy	7	87.5%
-resection	1	12.5%
**Delayed surgery**	5	62.5%
bile ducts removed and reconstruction with a Roux-en-Y loop	3	
pancreatoduodenectomy and total hepatectomy	1	
the gallbladder with cystic duct removed	1	
**Resection**	6	75%
R0	5	
R1	0	
R2	1	
**Radiotherapy**		
yes	4	50%
-after initial surgery	2	
-relapse	2	
no		50%
**Outcome**		
alive	6	75%
DOD	2	25%

**Table 3 cancers-16-03110-t003:** Reported series with more than 8 cases of biliary tract RMS. COG—the Children’s Oncology Group; AIEOP-STSC—l’Associazione Italiana di Ematologia e Oncologia Pediatrica—Soft Tissue Sarcoma Committee; EpSSG—the European Soft-Tissue Sarcoma Group; CWS—the Cooperative Weichteilsarkom Studiengruppe.

Case Series More than 8 Cases	No of Cases	Trial	Overall Survival	Main Conclusions
Spunt et al., 2000 [17]	25	COG	66%	Outcome of biliary RMS good despite residual disease, chemotherapy responsible for improving survival, primary aggressive surgery not recommended
Perruccio et al., 2017 [19]	10	AIEOP-STSC	50%	Multimodal therapy needed for biliary RMS
Guerin et al., 2019 [11]	30	EpSSG	85%	No differences between influence of surgery and irradiation on local recurrence, relapse is related to tumor size
Urla et al., 2019 [2]	17	CWS	58%	Predictive factors for survival: age < 10 years and botryoid histology,
Aye et al., 2020 [3]	17	COG	76.5%	Low-risk therapy for biliary RMS resulted in suboptimal treatment

## Data Availability

Patients’ data is available in Table 1. Patients’ data is anonymized to protect their privacy and comply with ethical standards.

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
