# Peer review of "Rhabdomyosarcoma of the Biliary Tract in Children: Analysis of Single Center Experience"

_cancers, 2024, doi:10.3390/cancers16173110_

Round 1
Reviewer 1 Report
Comments and Suggestions for Authors
The manuscript provides a comprehensive overview of the management and outcomes of rhabdomyosarcoma (RMS) of the biliary tract in children, drawing from a single center's experience. It addresses critical aspects of the disease, including epidemiology, diagnosis, treatment, and outcomes, and emphasizes the importance of surgical intervention in achieving favorable long-term results.
Recommendations for Strengthening the Manuscript:
1. Discuss potential standardized treatment protocols for biliary tract RMS.
2. Include a comparative analysis of different treatment modalities to provide clearer guidance on best practices and optimize therapeutic strategies. Additionally, discuss innovative therapies, such as targeted therapies or immunotherapies, and compare their effectiveness with traditional treatments.
3.Elaborate on diagnostic challenges and emerging technologies, including advanced imaging techniques and biomarkers, to enhance early detection and accurate diagnosis of biliary tract RMS.
4. Address any genetic mutations or molecular markers associated with biliary tract RMS to offer insights into the disease’s etiology and inform potential targeted therapies.
5. Include a brief discussion of RMS in adults, highlighting differences in presentation, management, and outcomes compared to pediatric cases, as well as any notable sex differences.
6. Incorporate relevant diagnostic figures to visually represent key data and findings.
Author Response
Response to the reviewer 1
Thank you very much for your thorough review of our paper and remarks that you have formulated. Your remarks were fully endorsed, more, they contributed to refinement of our paper.We present our answers and manuscript revisions according to your comments, hopefully satisfying you.
Below are the reviewer’s comments and our responses, which have been incorporated into the publication where necessary.
- Discuss potential standardized treatment protocols for biliary tract RMS.
- Include a comparative analysis of different treatment modalities to provide clearer guidance on best practices and optimize therapeutic strategies. Additionally, discuss innovative therapies, such as targeted therapies or immunotherapies, and compare their effectiveness with traditional treatments..
- Address any genetic mutations or molecular markers associated with biliary tract RMS to offer insights into the disease’s etiology and inform potential targeted therapies.
Response for 1,2,4 points together:
The standard of care for Rhabdomyosarcoma of the biliary tract requires multimodality treatment, including chemotherapy, surgical resection, and/or radiation therapy. The standard chemotherapy protocols combine vincristine, actinomycin, and cyclophosphamide/ifosfamide. Surgery aims for completeresection ofremnant tumor, which can be challenging due to the location in the biliary tract. Radiation therapy may be employed postoperatively to target residual disease, especially if complete surgical resection is not achievable. The protocol is often adapted based on the tumor's stage, histological subtype, and patient factors, with ongoing clinical trials helping refine these strategies. Multidisciplinary teams are crucial in tailoring treatment to optimize outcomes while minimizing complications.
New treatments, such as molecular targeted drugs and immunotherapies, have shown superior efficacy and beneficial clinical outcomes as compared to standard treatments in selected childhood malignancies as neuroblastoma, non-Hodgkin lymphoma, low grade gliomas, melanoma, however benefits of new approaches in pediatric RMS are not yet available. Genomic profiling of childhood soft tissue sarcoma which will eventually help to elucidate and discover clinically meaningful biomarkers as of today is not routinely performed. Studies on the role of tumor mutational burden and its role as a biomarker in predicting response of soft tissue sarcomas to immune checkpoint inhibitors are discouraging since they have a low mutational tumor burden. Further studies of biomarkers predicting the response to molecular targeted drugs and immune checkpoint inhibitors are needed.
3.Elaborate on diagnostic challenges and emerging technologies, including advanced imaging techniques and biomarkers, to enhance early detection and accurate diagnosis of biliary tract RMS.
Due to the rarity of biliary tract RMS and its presentation with non-specific symptoms like jaundice and abdominal pain, early diagnosis is often challenging. The intricate anatomy of the biliary system complicates imaging interpretation and biopsy procedures, leading to delays in accurate diagnosis. Although the authors do not have experience in using positron emission tomography (PET) in the diagnosis of biliary tract RMS (PET was not available at our center when the described patients were being treated), it appears, based on available case reports where this diagnostic technique was used, that it could be a valuable tool for distinguishing biliary tract RMS from bile duct cysts or other entities in this site. The use of PET has been already applied in the diagnosis of biliary tract cancers and their metastases in adults. PET scans utilize a radioactive tracer, typically fluorodeoxyglucose (FDG), which accumulates in tissues with high metabolic activity, like malignant tumors. RMS, being a highly aggressive and metabolically active tumor, usually shows increased FDG uptake on PET scans, leading to a "hot spot" appearance. The possibility of performing PET was introduced at our center in 2022, and the last patient presented in our study had a suspected finding in MRI after surgery.
- Include a brief discussion of RMS in adults, highlighting differences in presentation, management, and outcomes compared to pediatric cases, as well as any notable sex differences.
Our study aimed to present RMS in the specific biliary tree location which is extremely rare in adults (casuistic). It would be difficult to make comparisons in both age groups since there is practically no adult data in the literature and that being a pediatric center we have no access to such patients to present our experience. We found in the literature only single cases in this location, all with fatal outcome. On the other hand if we were to make comparisons of RMS in children and adults regardless of the tumor location it would be out of the scope of our article.
- Incorporate relevant diagnostic figures to visually represent key data and findings.
We have included both diagnostic images of the patients and a table summarizing the patients’ characteristics and treatment details in the publication: Figures 1, 2, 3, 4 and Table 3. This graphical presentation will contribute to the clarity of the publication.
The file containing the article also includes the changes required by the second reviewer. All corrections are highlighted in yellow
We hope that the changes meet your expectations and that the manuscript is now suitable for publication. Please do not hesitate to contact us if any further adjustments are needed.
We appreciate your time and consideration.
Sincerely,
Dorota Broniszczak
Corresponding author
Reviewer 2 Report
Comments and Suggestions for Authors
In this study the authors review the outcomes of patients with rhabdomyosarcoma of the biliar tract. The review can have interest for other practitioners treating patients with this disease, as in a not frequent one. There are somethings that should be improved in the manuscript:
- Material and methods: The type of study should be indicated (retrospective, etc)
- Results: Table 1. Each column should have a title
- Discussion: It is too long. Some of the explanations about the different treatments could be included in the introduction
.
Author Response
Response to the reviewer 2
Thank you very much for your thorough review of our paper and remarks that you have formulated. Your remarks were fully endorsed, more, they contributed to refinement of our paper. We revised the manuscript according to your comments which hopefully will satisfy you.
Below are the reviewer’s comments along with our responses, which have been incorporated into the publication where necessary.
- Material and methods: The type of study should be indicated (retrospective, etc)
The publication is a retrospective case series describing a group of pediatric patients with biliary tract RMS.
- Results: Table 1. Each column should have a title
Each column had a title, but it may not have been very clearly indicated. We have now highlighted the column titles more prominently.
- Discussion: It is too long. Some of the explanations about the different treatments could be included in the introduction
We have moved several paragraphs with a general character from the discussion to the introduction, as suggested by the reviewer. The discussion had already been shortened; it was much longer before. The complex and individualized approach requires discussing many aspects, hence the discussion is extensive to cover every detail.
The file containing the article also includes the changes required by the second reviewer. All corrections are highlighted in yellow
We hope that the changes meet your expectations and that the manuscript is now suitable for publication. Please do not hesitate to contact us if any further adjustments are needed.
We appreciate your time and consideration.
Sincerely,
Dorota Broniszczak
Corresponding author